# Highly efficient recycling of polyester wastes to diols using Ru and Mo dual-atom catalyst

Minhao Tang[1,2,6], Ji Shen[3,6], Yiding Wang[1,2], Yanfei Zhao [1,2] ✉, Tao Gan[4], Xusheng Zheng[5], Dingsheng Wang [3] ✉, Buxing Han [1,2] & Zhimin Liu [1,2] ✉

The chemical recycling of polyester wastes is of great significance for sustainable development, which also provides an opportunity to access various oxygen-containing chemicals, but generally suffers from low efficiency or separation difficulty. Herein, we report anatase $TiO_2$ supported Ru and Mo dual-atom catalysts, which achieve transformation of various polyesters into corresponding diols in 100% selectivity via hydrolysis and subsequent hydrogenation in water under mild conditions (e.g., 160 °C, 4 MPa). Compelling evidence is provided for the coexistence of Ru single-atom and O-bridged Ru and Mo dual-atom sites within this kind of catalysts. It is verified that the Ru single-atom sites activate $H_2$ for hydrogenation of carboxylic acid derived from polyester hydrolysis, and the O-bridged Ru and Mo dual-atom sites suppress hydrodeoxygenation of the resultant alcohols due to a high reaction energy barrier. Notably, this kind of dual-atom catalysts can be regenerated with high activity and stability. This work presents an effective way to reconstruct polyester wastes into valuable diols, which may have promising application potential.

Plastics, permeating every aspect of our lives, present formidable environmental challenges as they generate prodigious quantities of wastes, casting a shadow upon ecological harmony[1–3]. In the context of a circular economy, the chemical recovery and recycling of plastic wastes play a vital role,which has attracted much attention in recent years[4]. Compared to the conventional waste management methods such as incineration and landfills[5,6], the chemical recycling provides effective solutions to solve the problems caused by plastic waste, and also offers some novel routes to produce chemicals[7]. For instance, the thermocatalytic hydrocracking of polyethylene has been explored as a means to generate alkanes with diverse carbon chain lengths[8,9]. Polyesters, a class of synthetic plastics linked with ester group -C(=O) OC-, share about one third market of plastics, and chemical recycling of the discarded polyesters have been widely investigated via various

approaches including hydrolysis, alcoholysis, aminolysis, and hydrogenolysis[10–17]. However, the challenges related to reactivity of polyester degradation and selectivity towards target products still remain in many cases.

Diols, such as ethylene glycol and 1,2-propandiol, are important chemicals widely applied in industrial productions and daily lives[18]. The production of diols typically involves complex chemical processes and high energy costs[19]. Therefore, exploring simple and environmentally friendly approaches to access diols is an important and intriguing topic. Given that polyesters are generally derived from condensation coupling between hydroxyl and carboxyl groups of monomers like diols and dicarboxylic acid or hydroxyl carboxylic acids, it is possible to produce diols through the hydrogenolysis of polyesters if the carbonyl group of ester group is selectivly

[1]Beijing National Laboratory for Molecular Sciences, CAS Laboratory of Colloid and Interface and Thermodynamics, CAS Research/Education Center for Excellence in Molecular Sciences, Center for Carbon Neutral Chemistry, Institute of Chemistry, Chinese Academy of Sciences, Beijing, China. [2]University of Chinese Academy of Sciences, Beijing, China. [3]Department of Chemistry, Tsinghua University, Beijing, China. [4]Shanghai Synchrotron Radiation Facility, Shanghai Advanced Research Institute, Chinese Academy of Sciences, Shanghai, China. [5]National Synchrotron Radiation Laboratory, University of Science and Technology of China, Hefei, Anhui, China. [6]These authors contributed equally: Minhao Tang, Ji Shen. ✉e-mail: lianyi302@iccas.ac.cn; wangdingsheng@mail.tsinghua.edu.cn; liuzm@iccas.ac.cn

hydrogenated into $CH_2\text{-}OH$[20]. For instance, polylactic acid (PLA) has been reported to be capable of being hydrogenolyzed into 1,2-propandiol at higher temperatures[21]. However, achieving exclusive access of diols from polyesters remains challenging due to the formation of carboxylic acids and/or other byproducts[22,23]. Therefore, the catalysts capable of achieving this goal are highly desirable.

In recent years, atomically dispersed metal catalysts have garnered significant attention for their high atom utilization efficiency and unique properties tuned by coordination environments[24,25]. Down-sizing precious metal particles into single-atom (SA) size is an effective strategy to improve atom utilization efficiency, ultimately reducing production costs and enhancing the sustainability of related industries[26–28]. More recently, bimetallic SA catalysts have been reported to exhibit enhanced performances compared to mono-metallic SA catalyst[29,30]. This enhancement can be attributed to the interaction between bimetallic components, which optimizes the electron distribution around the metal center and provides multiple active sites in tandem, thus leading to optimal adsorption conformations of reactants and favorable reaction energy barriers[31–33]. Nevertheless, the contribution of partially unpaired metal SA sites in catalysts are frequently disregarded.

In this work, we report Ru and Mo dual-atom catalysts (DACs), in which Mo induces the formation of Ru SAs and O-bridged Ru and Mo dual-atoms. These DACs with multiple sites could synergistically convert various polyesters to corresponding diols in the presence of $H_2$ in water under mild conditions (e.g., 160 °C, 4 MPa). The $TiO_2$ supported Ru and Mo dual-atom with Ru:Mo molar ratio around 4:1 and Ru loading of 3.76 wt% displayed the highest performance, which could achieve recycling of various polyesters into corresponding diols in 100% selectivity. The microscopic characterization confirms the unique structure of anatase $TiO_2$ supported Ru and Mo bimetallic catalyst with atomically dispersed Ru and Mo SA sites and O-bridged Ru and Mo dual atomic sites, and a strong electronic interplay between Ru and Mo atoms. The Ru sites activate $H_2$ to provide active H for hydrogenaton of carboxyl groups into C-OH groups, while Ru-O-Mo sites remain a high reaction energy barrier to suppress hydrodeoxygenation of C-OH. In addition, the used catalyst could be regenerated via heating treatment in $O_2$ atmosphere, and kept unchanged activity after being reused for 10 runs.

## Results

Considering that degradation of polyesters into diols may undergo hydrolysis to dicarboxylic acids and diols or hydroxyl carboxylic acids depending on the structures of the polymers, and subsequent hydrogenation of the resultant carboxylic acids into diols, dual functional catalysts are therefore required to be capable of catalyzing the hydrolysis of polyesters and the subsequent hydrogenation of carboxylic acids exclusively to diols without hydrodeoxygenation of diols. In this work, the anatase $TiO_2$ nanoparticle-supported Ru and Mo dual-atom catalysts ($Ru_xMo_y/TiO_2$) were prepared using the procedures as illustrated in Supplementary Fig. 1. Typically, $RuCl_3$ and ammonium molybdate are dissolved in distilled water with dispersed $TiO_2$ nanoparticles, followed by addition of fresh $NaBH_4$ aqueous solution, resulting in the formation of $TiO_2$ supported Ru and Mo bimetal nanoparticles, which are further treated in air at 200 °C to access the desired catalysts. A series of $Ru_xMo_y/TiO_2$ with different Ru:Mo molar ratios were fabricated, and the catalyst with the Ru:Mo molar ratio of 4:1 (i.e., x:y = 4:1) and the Ru loading of 3.76 wt% is designated as $Ru_4Mo_1/TiO_2$, while the others are subscripted to indicate the feed amounts with the actual loadings measured by inductively coupled plasma optical emission spectrometer (ICP-OES, Table S1). In addition, the $TiO_2$ supported Ru with loading of 6.0 wt% ($Ru_6/TiO_2$) and Mo-doped $TiO_2$ nanoparticles with Mo loading of 1.74 wt% ($Mo_2/TiO_2$) were prepared as well for comparison.

The activities of the resultant catalysts were examined in catalyzing depolymerization of PLA in the presence of $H_2$ in water (Fig. 1a). It was indicated that the $TiO_2$ support and $Mo_2/TiO_2$ were effective only for the hydrolysis of PLA to lactic acid (LA) (Fig. 1a). The $Ru_6/TiO_2$ and $Ru_xMo_y/TiO_2$ catalysts showed activity for transformation of PLA to 1,2-propanediol. Especially, the $Ru_xMo_y/TiO_2$ catalysts with low ratios of Ru to Mo (e.g., $Ru_3Mo_{10}/TiO_2$, $Ru_2Mo_1/TiO_2$, $Ru_4Mo_1/TiO_2$) afforded 1,2-propanediol as the sole hydrogenated product, with its yields increase along with the Ru:Mo ratios of the catalysts, approaching 100% in the case of using $Ru_4Mo_1/TiO_2$ as the catalyst. The catalysts $Ru_6/TiO_2$ and $Ru_4Mo_{0.1}/TiO_2$ with a high Ru:Mo ratio also achieved transformation of PLA under the same other conditions, however, affording more hydrogenated products including 1,2-propanediol, 1-propanol, isopropyl alcohol, ethane and methane (Supplementary Figs. 2 and 3), consistent with the reported reports[34]. In comparison, $Ru_4Mo_{0.1}/TiO_2$ offered more 1,2-propanediol than $Ru_6/TiO_2$. The above findings indicate that the introduction of Mo in the $Ru_xMo_y/TiO_2$ catalysts significantly influenced their catalytic activity for hydrogenation, and the presence of Mo is favorable to the formation of 1,2-propanediol.

How does Mo impact the Ru activity in $Ru_xMo_y/TiO_2$? Prior to answering this question, $Ru_4Mo_1/TiO_2$ as the catalyst was used to optimize the reaction conditions for decomposition of PLA. The PLA degradation underwent hydrolysis to LA and subsequent LA hydrogenation, which was remarkably influenced by reaction temperature and $H_2$ pressure (Supplementary Figs. 4–7). The PLA hydrolysis and LA hydrogenation could simultaneously occur at 100 °C, affording a combined yield of LA and 1,2-propanediol approximately 30%. As temperature increased, the total yields of LA and 1,2-propanol increased accordingly, with a higher yield of 1,2-propanediol than that of LA. At 160 °C and a suitable $H_2$ pressure (e.g., 4 MPa), LA disappeared, and the yield of 1,2-propanediol approached 100%, suggesting that the generated LA from PLA hydrolysis was completely converted into 1,2-propanediol. However, further increasing temperature and pressure led to a slight decline in 1,2-propanediol yield caused by formation of byproducts including propanol and isopropanol (Supplementary Figs. 4 and 6). The stability of the $Ru_4Mo_1/TiO_2$ catalyst was examined in PLA decomposition at 160 °C and 4 MPa, and the used catalyst could be regenerated via being treated in an $O_2$ atmosphere at 200 °C, restoring its activity to a level similar to that of the fresh catalyst (Fig. 1b).

Taking the optimized conditions in hand, $Ru_4Mo_1/TiO_2$ was applied in degradation of PLA in a large scale, and a high yield (97%) of 1,2-propanediol was obtained, meanwhile it was also very effective for recycling the PLA straw and lid into 1,2-propanediol in high yields of 98% and 97% under the experimental conditions, respectively. Inspired by these results, $Ru_4Mo_1/TiO_2$ was employed in reconstruction of other polyesters including polyglycolic acid (PGA), poly (butylene succinate) (PBS), polycaprolactone (PCL), poly(1,4-butylene adipate) (PBA) and polybutylene adipate terephthalate (PBAT; Fig. 1c). Similar to PLA, PGA, and PCL were selectively transformed into corresponding ethylene diol and 1,6-hexanediol, respectively, in selectivity of 100%. The degradation of PBS generated 1,4-butanediol as the sole product, while PBA degradation produced 1,6-hexanediol and 1,4-butanediol. These findings demonstrate that the carboxylic acids derived from hydrolysis of polyesters were hydrogenated into corresponding diols, while the hydrodeoxygenation of diols were efficiently inhibited.

Furthermore, hydrogenation of ethyl acetate and molten PLA over $Ru_4Mo_1/TiO_2$ in the absence of water were performed. It was indicated that ethyl acetate could be completely hydrogenated into ethanol, and molten PLA into 1,2-propanediol in a yield (95%) analogous to that obtained in the water-based system. These results indicate that $Ru_4Mo_1/TiO_2$ can selectively catalyze hydrogenolysis of ester group into two hydroxyl groups. However, for PBAT decomposition, only hydrogenation of adipic acid occurred, while hydrogenation of

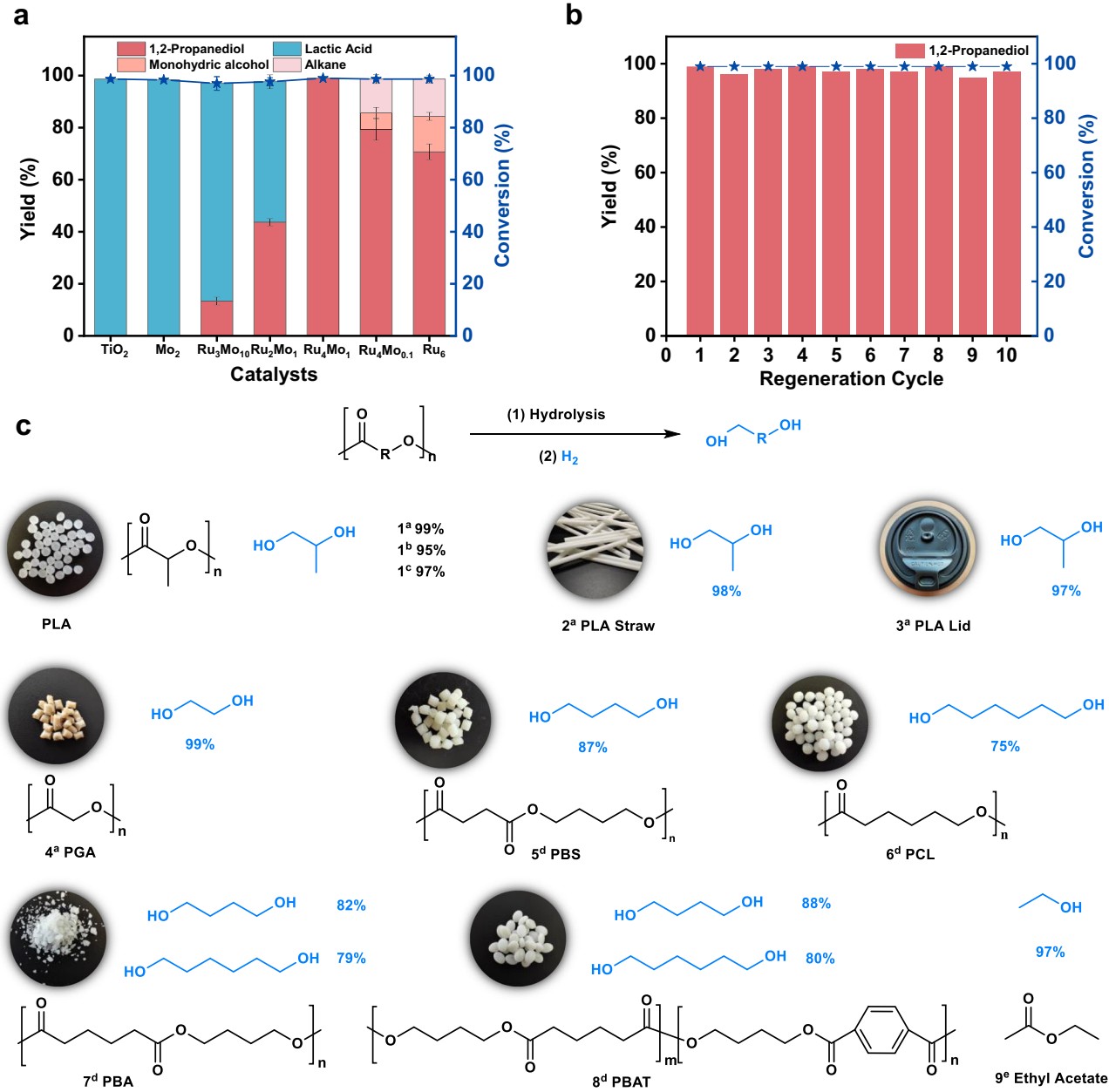

**Fig. 1 | Catalyst performances. a** Catalyst activity for PLA degradation.
**b** Regeneration stability of $Ru_4Mo_1/TiO_2$. **c** Hydrogenation of various ester samples.
Reaction conditions: [a]PLA, 72 mg; catalyst, 10 mg; $H_2O$, 0.5 mL; $H_2$, 4 MPa; 160 °C,
12 h. [b]Solvent-free, 180 °C. [c]Substrate, 1 g; $Ru_4Mo_1/TiO_2$, 100 mg; $H_2O$, 3 mL; $H_2$,
8 MPa. [d]48 h. [e]Solvent-free; substrate, 3 mmol; $Ru_4Mo_1/TiO_2$, 30 mg; $H_2$, 8 MPa. The
yield of each diol was determined by $^1H$ NMR analysis based on its theoretical yield
of 100%.

terephthalic acid did not proceed under the experimental conditions.
This highlights the remarkable ability of $Ru_4Mo_1/TiO_2$ for selectively
catalytic hydrogenation of aliphatic carbonyl into C-OH group.

## Characterization of the catalysts

To reveal the relationship between the structure of the catalysts and
their activity, the as-prepared $Ru_6/TiO_2$, $Mo_2/TiO_2$, and $Ru_xMo_y/TiO_2$
catalysts were characterized by means of different techniques. High-
resolution transmission electron microscopy (HR-TEM) observation
indicates that the $TiO_2$ support exhibited in the form of nanoparticles
(NPs) with an average size of 20 nm in all samples (Fig. 2a and Sup-
plementary Figs. 8–11). The aberration-corrected high-angle annular
dark-field scanning transmission electron microscopy (AC HAADF-
STEM) images provide more detailed information on the distribution
of Ru and Mo atoms in these samples. No noticeable NPs were

observed in $Ru_4Mo_1/TiO_2$ and $Mo_2/TiO_2$; instead, the Ru and Mo atoms
were observed to be present as SAs and incorporated into the defects
of the $TiO_2$ lattices. Energy dispersion spectroscopy (EDS) mapping
images display the uniform dispersion of Ru and Mo throughout the
entire $TiO_2$ support (Fig. 2b). Bright atomic columns in the AC HAADF-
STEM images were identified as Ru and Mo atoms (Fig. 2c), further
confirmed by the intensity profile along the dashed rectangles in
Fig. 2d[27,35,36]. This means that there exist many O-bridged Ru and Mo
(Ru-O-Mo) DA sites as well as Ru SAs in $Ru_4Mo_1/TiO_2$. To provide a
more comprehensive illustration of the formation of Ru-O-Mo DAs and
Ru SAs, we employed AC HAADF-STEM to observe $Ru_4Mo_1/TiO_2$ from
multiple angles and locations, and the collected AC HAADF-STEM and
EDS line-scan analysis images (Supplementary Figs. 12–14) provide
solid evidence of coexistence of Ru SAs and Ru-Mo DAs. Given the
discernible variation in activities of the catalysts with varying Ru:Mo

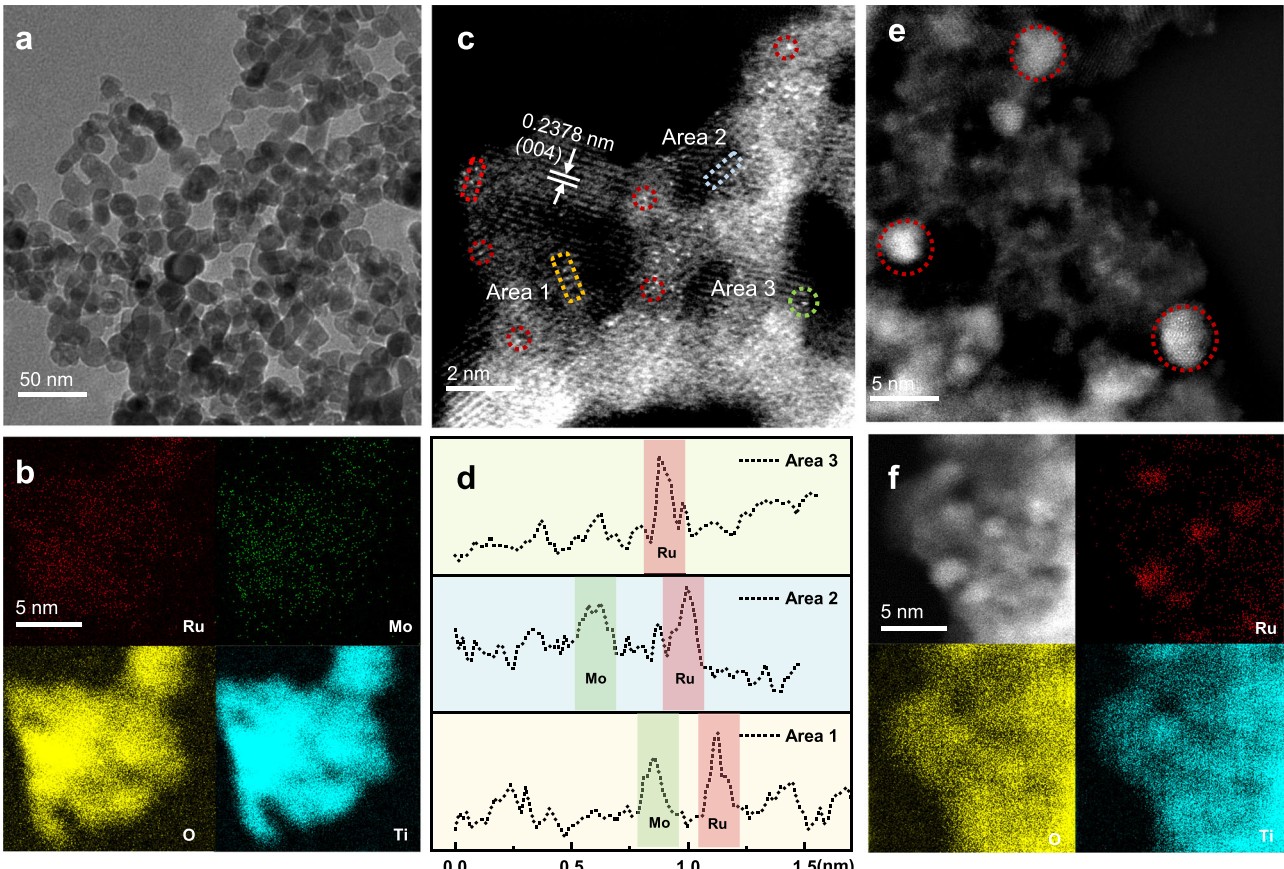

**Fig. 2 | Characterization of the catalysts. a** TEM image, (**b**) EDS mapping images and (**c**) AC HAADF-STEM image of $Ru_4Mo_1/TiO_2$. **d** The corresponding intensity profile of the parent $TiO_2$ along the dashed rectangles, showing the arrangement of the isolated Ru and Mo atoms. **e** AC HAADF-STEM and (**f**) EDS mapping images of $Ru_6/TiO_2$. In the EDS images, red represents for Ru, green for Mo, yellow for O, and cyan for Ti.

ratios for the PLA degradation, it is reasonable to infer that the Ru SA and Ru-O-Mo sites play divergent roles in the reaction process.

Focusing on the $TiO_2$ support of $Ru_4Mo_1/TiO_2$, the well-defined lattice fringes with a spacing of 0.2378 nm were observed, corresponding to the (004) planes of $TiO_2$. The absence of information on the Ru clusters/nanoparticles implies the possible presence of Ru atoms in the form of SAs in this sample. In contrast, the AC HAADF-STEM image and EDS mapping images of $Ru_6/TiO_2$ show that the Ru NPs with an average size of 3 nm are distributed on the surface of the $TiO_2$ support (Fig. 2e, f). From the above findings, it can be inferred that the Mo atoms in $Ru_4Mo_1/TiO_2$ may induce the atomic dispersion of Ru atoms due to the electronic interaction between the Ru and Mo atoms, which may be responsible for its catalytic activity[37,38]. The X-ray diffraction (XRD) patterns of the as-prepared catalysts only displayed characteristic peaks associated with anatase $TiO_2$ (Supplementary Fig. 15), indicating that the presence of Ru and Mo SAs hardly impacts the crystalline structure of the $TiO_2$ support.

As mentioned above, we observed a gradual deactivation of $Ru_4Mo_1/TiO_2$ as it was used in catalyzing PLA degradation under the experimental conditions (Supplementary Fig. 16). AC HAADF-STEM observation reveals the change of Ru SAs into clusters under $H_2$ atmosphere during the reaction process (Supplementary Fig. 17). Interestingly, the regenerated catalyst exhibited similar microstructure to the fresh one, with uniform dispersion of Ru and Mo dual-atoms (Supplementary Fig. 18), which is responsible for its almost unchanged activity. This corroborates the effectiveness of the regeneration approach for the used catalyst and underscores its potential for maintaining the catalytic performance of the system[39].

To acquire insight into the electronic structures and chemical environment of Ru and Mo atoms and to determine the oxidation states of Ru and Mo in the catalysts, we conducted X-ray absorption fine structure (XAFS) spectroscopy and X-ray absorption near edge structure (XANES) spectroscopy, respectively. As shown in Fig. 3a, the rising edges of the $Ru_4Mo_1/TiO_2$ and $Ru_6/TiO_2$ spectra lay between those of Ru foil and $RuO_2$, indicating that the valences of Ru in both catalysts are in the range of 0 ~ +4. The presence of the Mo atoms in $Ru_4Mo_1/TiO_2$ renders the Ru atoms to have a higher valence than that in $Ru_6/TiO_2$, ascribing to their electronic interaction. The Ru K-edge XANES spectra recorded for $Ru_xMo_y/TiO_2$ with different Ru:Mo molar ratios (Supplementary Fig. 19) show that the near-edge absorption energies ($E_0$) of all these samples are located between those of $RuO_2$ and the Ru foil, suggesting the positively charged feature of Ru SAs in these samples. Along with the Mo:Ru ratio increase in the samples, a distinct shift of the absorption edge toward higher energy was observed, indicating the stepwise increased oxidation state of Ru atoms in these samples. The normalized XANES spectra of Mo K-edge showed a decrease in the valence of Mo in $Ru_4Mo_1/TiO_2$ compared to that in $Mo_2/TiO_2$, corresponding to the increase in Ru valence, which also supports the strong electronic interaction between Ru and Mo atoms (Fig. 3b).

To investigate the surface oxygen atoms of the $Ru_4Mo_1/TiO_2$ catalyst, soft X-ray absorption spectroscopy (XAS) analysis was performed, which indicates that the absorption intensity of the edge-front peak of $Ru_4Mo_1/TiO_2$ decreased compared to that of $Ru_6/TiO_2$, suggesting an elevated valence state of surface oxygen in $Ru_4Mo_1/TiO_2$ (Supplementary Fig. 20). The $K^2$-weight extended X-ray absorption fine

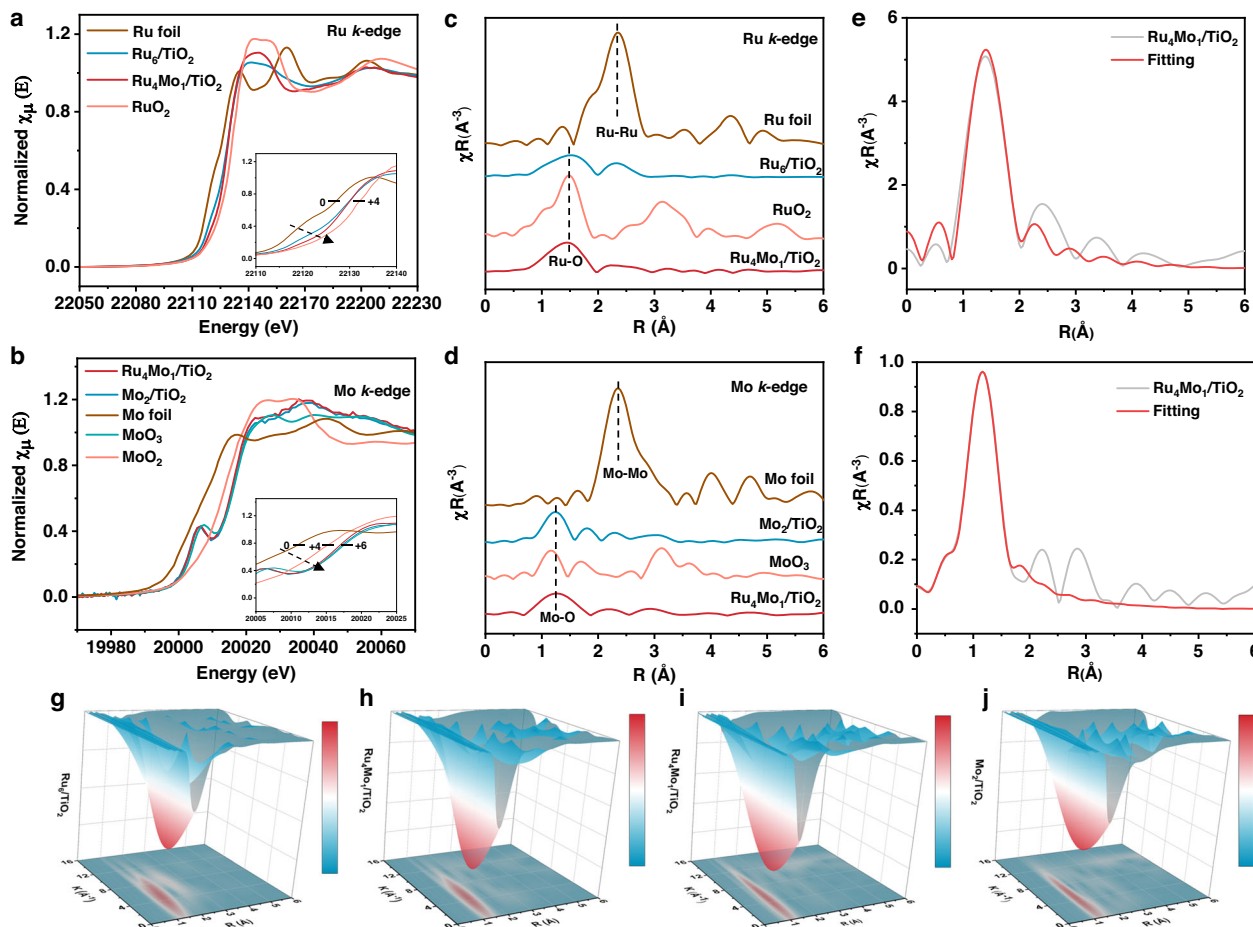

**Fig. 3 | Structural characterizations by XAFS.** Normalized XANES spectra: (**a**) Ru K-edge, inset: absorption edges of Ru; (**b**) Mo K-edge, Inset: absorption edges of Mo. *R*-space EXAFS spectra: (**c**) Ru samples, (**d**) Mo samples. EXAFS fitting: (**e**) Ru, (**f**) Mo of Ru-O-Mo. Inset: schematic diagram of the $Ru_4Mo_1/TiO_2$ structure. 3D contour maps of WT-XAFS spectra: (**g**) Ru of $Ru_6/TiO_2$, (**h**) Ru and (**i**) Mo of $Ru_4Mo_1/TiO_2$, and (**j**) Mo of $Mo_2/TiO_2$.

structure (EXAFS) spectra of the samples in *k*-space and *R*-space by Fourier transform were collected to provide the bonding information (Fig. 3c and Supplementary Fig. 21). The Fourier-transformed R-space spectrum of $Ru_4Mo_1/TiO_2$ showed only one peak at 1.5 Å, attributing to the first-shell Ru-O path, while those of both Ru foil and $Ru_6/TiO_2$ showed peaks at 2.5 Å, ascribing to the Ru-Ru path. These results confirm that the Ru atoms are present in the form of individual atoms in $Ru_4Mo_1/TiO_2$. The spectra of other $Ru_xMo_y/TiO_2$ are shown in Supplementary Fig. 22, most of which has no obvious peak assigning to Ru-Ru path, excluding the presence of Ru clusters or NPs. However, the spectrum of $Ru_5Mo_{0.1}/TiO_2$ displayed a small peak to Ru-Ru path, indicating the presence of Ru clusters or NPs in this sample, which may be responsible for its performance different from that of $Ru_4Mo_1/TiO_2$.

Similarly, the $K^2$-weight EXAFS and the *R*-space spectra of Mo in $Ru_4Mo_1/TiO_2$ and in $Mo_2/TiO_2$ demonstrated a single peak at 1.3 Å attributing to the Mo-O path, but no significant peak at 2.4 Å ascribing to the Mo-Mo path, as observed in that of Mo foil (Supplementary Fig. 23 and Fig. 3d), which confirms the atomic dispersion of Mo in $Ru_4Mo_1/TiO_2$. Especially, compared with those of $RuO_2$ and $Mo_2/TiO_2$, the Ru-O and Mo-O peaks in the spectra of $Ru_4Mo_1/TiO_2$ were asymmetrical and slightly shifted, implying that the electronic interaction between of Ru and Mo atoms may affect the coordination environment of the Ru and Mo centers[40].

XAS characterization on the used and regenerated catalysts was conducted to complement the electron microscopy findings (Supplementary Figs. 24 and 25). Consistent with the AC HAADF-STEM results, the XAS analysis in *R*-space reveals the presence of Ru clusters and Mo

SAs in the used catalyst. However, in the regenerated catalyst no agglomerated Ru clusters were observed, and SAs instead were present. This means that the regeneration process of the used catalyst led to the formation of Ru and Mo SAs once again.

Furthermore, the IFEFFIT module of Artemis was used to fit *R*-space data, and the corresponding fitting parameters are shown in Fig. 3e, f, Table S2, and Supplementary Figs. 26–30. The EXAFS fitting results for $Ru_4Mo_1/TiO_2$ (Fig. 3e, f) indicate that Ru and Mo atoms are both coordinated with four O atoms in the first shell. In combination with the AC HAADF-STEM observation, it can be deduced that two adjacent Ru and Mo atoms may share one O atom in $Ru_4Mo_1/TiO_2$. That is, adjacent Ru and Mo individual atoms are bridged with O, i.e., in the form of Ru-O-Mo. Moreover, Wavelet transform (WT) contour maps provide complementary information, showing Ru-Ru scattering paths only in $Ru_6/TiO_2$, and no Mo-Mo scattering paths in $Mo_2/TiO_2$ and in $Ru_4Mo_1/TiO_2$, which also support the SA dispersion of Ru and Mo in $Ru_4Mo_1/TiO_2$ (Fig. 3g–j and Supplementary Fig. 31).

X-ray photoelectron spectroscopy (XPS) analysis further revealed the interfacial electronic interplay by showing the oxidation states of Ru and Mo atoms, which supports the presence of Ru and Mo SAs in the catalyst. The Ru $3d_{5/2}$ bonding energys (BEs) of $Ru_xMo_y/TiO_2$ increased with the Mo loadings in the samples, while the BEs of Mo $3d_{5/2}$ in these samples decreased accordingly (Supplementary Figs. 32 and 33), which agreed well with the calculated results. To reveal the binding environment of the Ti and Ru atoms in the as-prepared catalysts, electron paramagnetic resonance (EPR) analysis was performed since it is highly sensitive to the

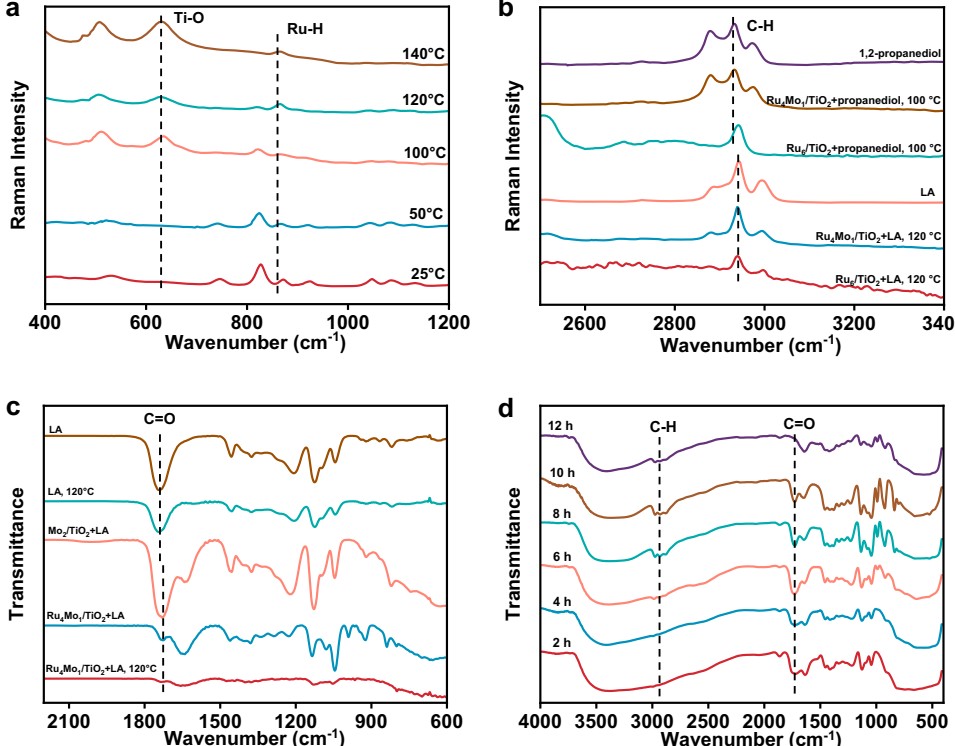

**Fig. 4 | Identification of the surface reaction species.** Operando Raman spectra: (**a**) $Ru_4Mo_1/TiO_2$ applied in the hydrogenation LA at different temperatures, (**b**) comparison of $Ru_4Mo_1/TiO_2$ and $Ru_6/TiO_2$ employed in hydrogenation of 1,2-propanediol at different temperatures. **c** Operando FTIR spectra of LA transformation over $Ru_4Mo_1/TiO_2$ and $Mo_2/TiO_2$ at room temperature and 120 °C in air. **d** Ex situ FTIR spectra of LA transformation over $Ru_4Mo_1/TiO_2$ for different time under optimal reaction conditions.

paramagnetic species containing unpaired electrons. It is demonstrated that the addition of Mo atoms in the samples caused a decrease in the $g$-value of the environment electrons of Ru (Supplementary Fig. 34). Notably, the EPR spetrum of $Ru_4Mo_1/TiO_2$ has a weaker signal than that of $Ru_6/TiO_2$, which suggests that the Ru SA was trapped into the lattice of $TiO_2$[41].

## Mechanistic studies

Considering that degradation of PLA over $Ru_4Mo_1/TiO_2$ in water underwent hydrolysis of PLA to LA and subsequent hydrogenation of LA to 1,2-propanediol, the interaction between $Ru_4Mo_1/TiO_2$ and $H_2$, and the hydrogenation of LA over $Ru_4Mo_1/TiO_2$ were monitored by operando Raman spectroscopy to explore the catalytic mechanism. As shown in Fig. 4a, a prominent peak appeared at 867 cm$^{-1}$ in the temperature-dependent Raman spectra of the catalyst under the $H_2$ atmosphere, assigning to the vibration of the Ru-H bond. Clearly, this peak appears even at 25 °C, and its intensity slightly increases with temperature, indicating that this catalyst has a strong ability to split $H_2$ in a wide temperature range (25–140 °C), which provides a basis for LA hydrogenation in the PLA degradation process.

From the operando Raman spectra shown in Fig. 4b, it is clear that $Ru_4Mo_1/TiO_2$ and $Ru_6/TiO_2$ show obviously different performances for hydrogenation of LA and 1,2-propanediol under the experimental conditions. In the spectrum collected for hydrogenation of propanediol over $Ru_4Mo_1/TiO_2$, the three bands around 2900 cm$^{-1}$ assigning to saturated C-H bonds of propanediol hardly changed compared to those of pure propanediol, reflecting $Ru_4Mo_1/TiO_2$ was ineffective for further reaction of propanediol, while that obtained over $Ru_6/TiO_2$ exhibits an obvious difference, indicating that $Ru_6/TiO_2$ could catalyze the further reaction of propanediol (Fig. 4b). These findings indicate that $Ru_4Mo_1/TiO_2$ can effectively inhibit the hydrodeoxygenation of

hydroxyl group of propanediol, which may be ascribed to its unique structure with coexistence of Ru SA sites and Ru-O-Mo sites.

To further probe the catalytic intermediates, operando Fourier-transform infrared spectroscopy coupled with surface-enhanced infrared absorption spectroscopy (FTIR-SEIRAS) were performed. As displayed in Fig. 4c, at both room temperature and 120 °C, the characteristic C=O vibration peak of LA at 1741 cm$^{-1}$ shifted to 1727 cm$^{-1}$ upon LA was adsorbed on both $Mo_2/TiO_2$ and $Ru_4Mo_1/TiO_2$ catalysts. This blue shift may be attributed to the interaction between the Mo sites of the catalysts and carbonyl O of LA, and suggests elongation of the C=O bond and an increase in the vibration frequency. As depicted in Fig. 4d, the characteristic C=O peak of LA at 1727 cm$^{-1}$ gradually became weak with the reaction time and almost disappeared as the reaction proceeded for 12 h, meanwhile and the C-H bond peaks in fatty alcohols emerged around 2930 cm$^{-1}$, providing the evidence on the transformation of carbonyl to -C-OH group over the $Ru_4Mo_1/TiO_2$ catalyst[42]. These findings, presented in Fig. 4, provide valuable insights into the catalytic behavior and intermediates involved in the hydrogenation of LA over $Ru_4Mo_1/TiO_2$ catalyst in aqueous conditions.

Considering that $Ru_4Mo_1/TiO_2$ can realize selective hydrogenation of LA to 1,2-propanediol, but inhibit its further hydrodeoxygenation to isopropanol or N-propanol, we conducted theoretical calculations to reveal the catalytic mechanism of $Ru_4Mo_1/TiO_2$ for selective LA hydrogenation to 1,2-propanediol. The spectroscopic results provided insights into the Ru-O-Mo configuration of the active site, based on which some initial adsorption configurations of LA were first screened (Supplementary Figs. 35 and 36). The most stable adsorption configuration of LA at Ru-O-Mo is illustrated in Fig. 5a. Notably, LA molecule is preferentially adsorbed at the Mo site rather than Ru site via coordination with C=O of LA, with an adsorption energy of −1.76 eV.

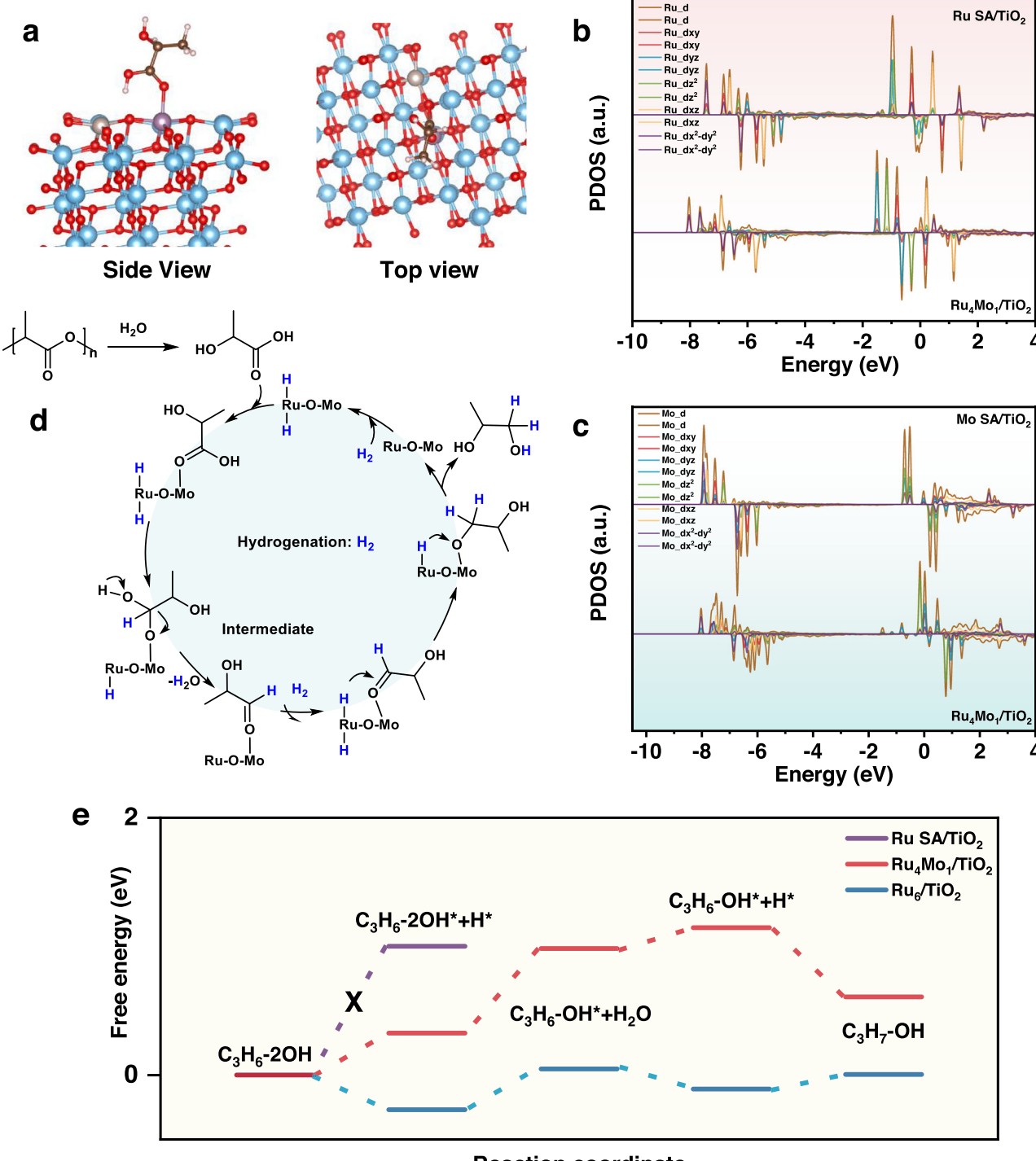

**Fig. 5 | Reaction paths and electronic structure of catalysts. a** Schematic diagram of the optimized adsorption structure of LA at the Ru-O-Mo site. (Gray: Ru, purple: Mo, red: O, blue: Ti and pink: H). The atomic coordinates of the optimized computational models are shown in Supplementary data 1. PDOS of (**b**) Ru *4d* of Ru SA/TiO₂ and Ru₄Mo₁/TiO₂ and (**c**) Mo *4d* of Ru₄Mo₁/TiO₂ and Mo₂/TiO₂. **d** Proposed reaction route of PLA recycling. **e** Free energy diagrams of 1,2-propanediol hydrogenation to isopropanol over different catalysts.

To further elucidate the role of the Ru-O-Mo active site in the hydrogenation of LA, we predicted the electronic structures of the Ru and Mo atoms. Figures 5b, c and Supplementary Figs. 37–39 show the projected density of states (PDOS) of Ru, Mo *4d* and O *2p* orbitals in Ru-O-Mo, Ru SA, Mo SA, and Ru₆/TiO₂, respectively. Obviously, comparing with Ru SA and Mo SA, the formation of Ru-O-Mo configuration induces significant changes in the distribution of 4d orbitals at −8 to −5

and −2 to 2 eV as well as approximately 2.5 eV relative to the Fermi level. Particularly, the modifications in $d_{xy}$ and $d_{yz}$ orbital states demonstrate the strong *d-d* interaction within the site plane, resulting in decreased orbital energy and enhanced electron delocalization, which may inhibit the hydrodeoxygenation of 1,2-propanediol.

Additionally, the distribution of *4d* orbitals of metal atoms in Ru-O-Mo with the LA adsorption at different metal sites was investigated

(Supplementary Fig. 39). It is indicated that the $4d$ orbital distribution of Mo in the Mo SA and in Ru-O-Mo site after adsorbing LA is similar, indicating that the Mo atom in the Ru-O-Mo site maintains the similar role to that of Mo SA. For the Ru SA site, the distribution of $4d$ orbitals before and after the LA adsorption remains unchanged, reflecting that Ru SA hardly adsorbs LA, which is consistent with the calculation results. However, the Ru $d_{xy}$ orbital of Ru-O-Mo varies significantly between −2 and 1 eV after LA adsorption, indicating that Ru is also influenced by LA adsorption at the Mo site.

Furthermore, the Bader charges of the Ru and Mo in Ru-O-Mo before and after LA adsorption were calculated. The Bader charges of Mo in Ru-O-Mo decrease from −2.20 to −2.27, while those of carbonyl O in LA increase from 1.07 to 1.15 as LA is adsorbed on Mo-O-Ru. This suggests that the adsorption of LA at the Mo-O-Ru site makes the Mo atom lose more electrons, along with carbonyl O in LA gaining more electrons. Notably, the Bader charges of Ru only increase from −1.49 to −1.48, suggesting little influence on its electronic structure after LA adsorption. The above findings demonstrate that the adsorption of LA at the Ru-O-Mo site results in the redistribution of electrons, with electron transfer from Mo to carbonyl O site. Consequently, the electron transfer pathway between Mo and O facilitates the activation of the C=O bond adsorbed at the Mo atom, thereby lowering the reaction energy barrier.

Based on the experimental results and discussion presented above, we propose a reasonable reaction pathway for the depolymerization of PLA (Fig. 5d). At temperatures higher than 100 °C, PLA undergoes hydrolysis catalyzed by weakly acidic $TiO_2$ support, resulting in the formation of LA. Simultaneously, $H_2$ dissociates, and hydrogen species spillover from the Ru SA sites to oxygen vacancies of $TiO_2$, facilitating subsequent hydrogenation. The generated LA molecules adsorb onto Ru-O-Mo sites via coordination between Mo and carbonyl O. The dehydration reaction takes place initially, and the formed unstable aldehyde intermediate rapidly undergoes subsequent hydrogenation, leading to the formation of 1,2-propanediol. Due to the weak adsorption at the Ru-O-Mo sites, the generated 1,2-propanediol molecules can easily desorb from $Ru_4Mo_1/TiO_2$, avoiding further hydrogenation.

Figure 5e illustrates the energy barrier of each step for hydrogenation of 1,2-propanediol to isopropanol over different catalysts. Obviously, the step $C_3H_6-2OH^* + H^* \longrightarrow C_3H_6\text{-}OH^* + H_2O$ over $Ru_4Mo_1/TiO_2$ has a higher energy barrier (0.66 eV), making it an inefficient hydrogenation pathway. In comparison, $Ru/TiO_2$ possesses bulk active sites, enabling the adsorbed 1,2-propanediol to react with active hydrogen until complete conversion into alkanes (Supplementary Figs. 40 and 41). In contrast, the constructed Ru SA/$TiO_2$ fails in 1,2-propanediol adsorption, implying that the Ru SAs in $Ru_4Mo_1/TiO_2$ has the sole capability for hydrogen activation.

## Discussion

The present study reports a kind of Ru and Mo dual-atom catalysts with Ru SA and O-bridged Ru and Mo dual-atom sites, and demonstrates their effectiveness for the degradation of polyesters into diols in the presence of hydrogen in water under mild conditions. From the findings of PLA degradation over $Ru_xMo_y/TiO_2$, it can be concluded that the degradation of PLA undergoes hydrolysis of PLA into LA and the subsequent hydrogenation of LA into 1,2-propanediol. The Ru SA sites or Ru NPs in the catalysts activate $H_2$ to provide active H* for LA hydrogenation to 1,2-propanediol, and the Ru-O-Mo sites inhibit further hydrogenation of 1,2-propanediol due to the high reaction energy barrier and weak adsorption for it. However, the Ru NPs in the catalyst can catalyze the hydrodeoxygenation of 1,2-propanediol to alkanes. Therefore, the disparity in activity of the $Ru_xMo_y/TiO_2$ catalysts for PLA degradation as shown in Fig. 1a can be explained as follows. For $Ru_xMo_y/TiO_2$ with higher Ru/Mo ratio without Ru NPs (e.g., $Ru_4Mo_1/$

$TiO_2$), more Ru SA sites are present, promoting activation of $H_2$ to provide more active H* for hydrogenation of LA, meanwhile there exist enough Ru-O-Mo sites to inhibit hydrodeoxygenation of 1,2-propanediol, thereby rationalizing higher catalytic activity associated with higher Ru:Mo ratios and affording diols in 100% selectivity. For the $Ru_xMo_y/TiO_2$ catalysts with Ru NPs (e.g., $Ru_4Mo_{0.1}/TiO_2$), the hydrodeoxygenation of 1,2-propanediol cannot be inhibited, thus affording diols in declined selectivity.

This work offers a kind of robust Ru and Mo dual-atom catalysts for production of diols from waste polyester plastics. The findings, presented in a comprehensive manner, contribute to advancement in chemical recycling of polyesters and synthesis of diols.

## Methods

### Reagents

Titanium dioxide (anatase, analytical grade, Aladdin), ruthenium chloride trihydrate ($RuCl_3 \cdot 3H_2O$, analytical grade, Adamas), ammonium molybdate ($H_{24}Mo_7N_6O_{24} \cdot 4H_2O$, analytical grade, J&K Scientific), sodium borohydride ($NaBH_4$, analytical grade, J&K Scientific), ethyl acetate ($C_4H_8O_2$, guaranteed grade, Concord Technology) and DL-Lactic Acid ($C_3H_6O_3$, >85.0%, TCI) were used. A variety of plastics are obtained directly from the corresponding production industry, PLA straw and PLA lid are wastes after use without further treatment. All the reagents and absolute ethanol were used without further purification. The distilled water is boiled for 1 min for degassing treatment.

### General procedures for preparation of catalysts

50 mg of titanium dioxide nanoparticles and 50 mL of distilled water were added to a 100 mL three-necked bottle and dispersed by high-energy ultrasound to form a uniform suspension. Then, a certain dose of bimetallic precursor solution with desired metal molar ratios (or a certain dose of monometallic precursor solution) was added to the titanium dioxide suspension and mixed well. Next, an excess of $NaBH_4$ solution was added drop by drop under high-energy ultrasonication. After filtration and being washed for 3 times, the resultant solid sample was dried in a vacuum oven at 60 °C. The dried catalyst was placed in a flowing oxygen atmosphere and heated at 200 °C for 3 h.

### Regeneration of the used catalysts

After being separated from the reaction solution, the used catalyst was washed for 3 times and dried in a vacuum oven at 60 °C. Then it was placed in a flowing oxygen atmosphere and heated at 200 °C for 3 h. Subsequently, the regenerated catalyst was reused for the next run.

### EXAFS analysis

Mo, Cu K edge X-ray absorption fine structure experiments were carried out on the BL11B beam line of the Shanghai Synchrotron Radiation Facility. Data processing and analysis were performed following standard methods. The extended X-ray absorption fine structure data were isolated using the Athena software. The $E_0$ values of 20000 and 22117 eV were used to calibrate all data with respect to the first inflection point of the absorption K-edge of Mo and Ru foil. The backscattering amplitude and phase shift functions for specific atom pairs were calculated ab initio using the FEFF8 code. X-ray absorption data were analyzed using standard procedures, including pre- and post-edge background subtraction, normalization with respect to edge height, Fourier transformation and non-linear least-squares curve fitting. The normalized $k^2$-weighted EXAFS spectra, $k^2 \times (k)$, were Fourier transformed in a k range from 1.4 to 11.0 $Å^{-1}$, to evaluate the contribution of each bond pair to the Fourier transform peak. The experimental Fourier spectra were obtained by performing an inverse Fourier transformation with a Hanning window function with r between 1.0–3.0 Å. The $S_{02}$ (amplitude reduction factor) value was determined based on the fitting results of metal-foil.

## General procedures for the degradation of polyesters over $Ru_4Mo_1/TiO_2$

The degradation reaction was performed in a stainless-steel autoclave equipped with a Teflon inner tube (15 mL inner volume) and a magneton. The desired amounts of polyester, catalyst and water were loaded into the autoclave, and $H_2$ was then charged into the autoclave after being degassed at room temperature. Subsequently, the autoclave was moved into a furnace set at the desired temperature (e.g., 160 °C), and taken out to cool naturally after desired reaction time. The liquid products were verified by $^1H$ and $^{13}C$ NMR spectroscopy, and their yields were determined based on the $^1H$ NMR spectra. Chemical shifts were given in ppm relative to tetramethylsilane (TMS). The gaseous products were determined by GC-MS. Paraformaldehyde was employed as the internal standard for the $^1H$ NMR analysis. The moles of paraformaldehyde were calculated based on its mass and molecular weight. The amounts of target products were determined by the following formula.

$$n_i = \frac{n_q \times a_i \times z_i}{6a_q} \quad (1)$$

where $n_i$ is the mole of each product, $n_q$ is the mole of paraformaldehyde, $a_i$ is the $^1H$ NMR characteristic peak area of each product, $a_q$ is the $^1H$ NMR peak area of paraformaldehyde, $z_i$ is the H numbers of characteristic $^1H$ NMR peak for each product.

The conversion of polyester was calculated as follows:

$$conversion = \frac{m_0 - m_s}{m_0} \times 100\% \quad (2)$$

where $m_0$ is the feeding mass of polyester and $m_s$ is the weight of solid residue after reaction.

The yield of each product or byproduct, for example, the products and byproducts from PLA degradation including lactic acid, 1,2-propanediol, 1-propanol, isopropyl alcohol, and propane was calculated as

$$yield = \frac{n_i}{n_0} \times 100\% \quad (3)$$

where $n_i$ is the mole of each product and $n_0$ represents the moles of the structural units of the feeding plastics.

The selectivity of each product or byproduct was calculated as follows:

$$selectivity = \frac{n_i}{\sum n_j} \times 100\% \quad (4)$$

where $\sum n_j$ is the overall moles of hydrogenation products.

All trials were repeated at least 3 times to ensure accuracy, and the mean standard deviation of reported results was <5%. All products were quantified using the internal standard method.

## DFT calculations

We performed density functional theory calculations using the Vienna ab initio simulation package (VASP)[43]. The projector augmented wave method and generalized gradient approximation of Perdew, Burke, and Ernzerh of (GGA-PBE) were applied to describe the exchange-correlation functionals[44]. To prevent erroneous interactions between neighboring periodic images, a vacuum layer of 12 Å along the z-axis was added. The models were fully relaxed to reach a convergence criterion of $10^{-5}$ eV in total energy and 0.02 eV Å$^{-1}$ in residual forces. The cutoff energy was set to 450 eV. A $2 \times 2 \times 1$ Monkhorst-Pack k-point mesh was adopted for all calculations[45]. To better describe the van der Waals force in these models, semi empirical DFT-D3 dispersion

correction (3-body terms) was adopted[46]. The pre- and post-processing data were produced by VASPKIT[47]. The atomic structures were analyzed by using the VESTA code[48].

## Data availability

All data supporting the findings of this work are available within the paper and its Supplementary Information. Source data are provided with this paper.

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

## Acknowledgements

This work was financially supported by the National Natural Science Foundation of China (22233006, L, 22121002, L and 22325101, W) and the Chinese Academy of Sciences (027GJHZ2022053MI, L). We thank the BL11B station of Shanghai Synchrotron Radiation Facility (SSRF) and the BL10B Station of National Synchrotron Radiation Laboratory (NRSL) measurement for providing beam time to support this work.

## Author contributions

Z.L., D.W., and Y.Z. designed the project and prepared the manuscript for publication. M.T. and J.S. carried out the experiments, collected the data and contributed to the writing of the manuscript. Y.W. carried out the calculations. T.G. and X.Z. performed and analyzed the XAS experiments. B.H. oversaw the project.

## Competing interests

The authors declare no competing interests.
