## [Peer Review File · Nature Communications]

REVIEWER COMMENTS

Reviewer #1 (Remarks to the Author):

The authors present a very intriguing work of decomposing polyester wastes to diols via using Ru and Mo dual-atom catalysts supported by TiO₂. The topic is of great environmental implications and the authors have achieved desirable experimental/practical results. However, there are some problems to be addressed before the manuscript can be acceptable for publication in Nature Communications.

1. Why do the yields in Figure 1b fluctuate slightly with the number of Regeneration Cycle increasing instead of a descending trend?
2. The authors points out that “the electron transfer from Mo to Ru sites”(in Line 302), i.e., Ru atoms act as electron acceptors, which seems to contradict with “The presence of the Mo atoms in Ru₄Mo₁/TiO₂ renders the Ru atoms to have a higher valence than that in Ru₆/TiO₂”(in Line 181-182),i.e., Ru atoms act as electron donators.
3. If “the Ru SA site has the sole capability for hydrogen activation”(in Line 318) is true, then it will be difficult to understand that “The Ru NPs in the catalyst can catalyze the hydroxydehydrogenation of 1,2-propanediol to alkanes”(in Line 327). That is to say, where is 1,2-propanediol adsorbed on Ru NPs if not Ru atoms?
4. In the sentence “where m_i is the mass of each product and m_0 represents the feeding mass of plastics.” (in Line 384), “mole” should be used instead of “mass”. The same is also true for “the overall mass of hydrogenation products” in Line 387.
5. The authors are encouraged to provide the adsorption energies of different adsorption configurations of LA on the Ru₄Mo₁/TiO₂ together with the configuration pictures in Supplementary Fig.29, Fig.30, Fig.34. and Fig.35.
6. The pictures mismatch letters captions in Supplementary Fig.33. (i.e., abcdef instead of adcebc)
7. The reaction energy barriers should be positive instead of negative, described below Supplementary Fig.35.
8. The authors are encouraged to provide the adsorption energies and also its dissociation energies of H₂ on different metal atoms on the catalysts to further substantiate the conclusion that “the Ru SA sites activate H₂ to offer H for hydrogenation reactions”.

Some language usage mistakes:

9. “X-ray adsorption fine structure (EXAFS)” in Line 194 should be “X-ray absorption fine structure (EXAFS)”.
10. “Supplementary Fig. 20-S24” (in Line 210) should be “Supplementary Fig. 20-24”.

Reviewer #2 (Remarks to the Author):

The authors in this work have developed an interesting Ru-Mo dual-atom catalyst which demonstrates outstanding catalytic performance in converting various polyesters into diols. The structures and roles of Ru SA and Ru-O-Mo sites have been illustrated clearly. However, there are a few points that need to be addressed before this manuscript can be published:

1. In this manuscript, diols are produced via hydrolysis and subsequent hydrogenation. However, carboxylic acid is normally more difficult to be hydrogenated than ester, more details, such as NMR spectra of reaction mixtures containing various intermediate products, should be supplemented to prove this route. It is not clear enough what internal standard is and how the yields of intermediate and final products are exactly determined, thus Methods and Supporting Information need to be more detailed in analytical procedures and calculations.
2. In Fig.2, the positions of b and c are disordered. In Fig.2d, it is not clear how the two atoms in Area 1 or 2 are determined to be Ru and Mo, respectively, EDS mapping cannot present atomic-resolution.
3. The sentence “.....of Ru6/TiO2 show that the Ru NPs with an average size of 3 nm are distributed on the surface of the TiO2 support, without Ru SAs”: The STEM image cannot prove it and it is unreasonable that no Ru SA distributes on this catalyst at all.
4. In Fig.4c and d, more information should be added including the co-feed of H2, catalyst and temperature.
5. In the caption, “PLA 1 mmol” is not accurate, it should be corrected as “PLA 72 mg”.
6. In Methods, the mass of products and feeding plastics is used to calculate yield and selectivity, but hydrogenated products are heavier than feeding plastics, for example, the mass of 1 mmol of propanediol is 76 mg, the mass of corresponding PLA is 72 mg. The yield and selectivity should be calculated based on the amount of carbon.

Reviewer #3 (Remarks to the Author):

In this manuscript, the authors demonstrated the efficient recycling of polyester wastes to diols using Ru and Mo dual atoms catalysis. This work is definitely a significant with respect to current global challenges and will have remarkable impact in the field. However, there are certain points to be addressed in a revision in order that this manuscript may fulfils the high standards of Nature Communications.

- The authors have used a combination of Ru and Mo, as Ru4Mo1 is the major catalyst among other combinations tried for Better conversion. It would be beneficial for the readers to understand the rationale behind this preference.
- Did the authors try a blank reaction with TiO2 to elaborate more on the role of TiO2 nanoparticles? How is it different when compared to TiO2 as support?
- While the authors claim the formation of DAs and SAs, the evidence provided by HAADF, STEM, and EDS spectra is not entirely convincing. Few other characterizations are recommended to support the fact. For e.g., STM analysis for further validating the fact that DAs and SAs were formed in the reaction.
- Exploring the reactivity when using Ru nanoparticles instead of single atoms would provide valuable insights into the catalytic mechanism.

- Although the catalyst exhibits excellent recyclability, it remains unclear how SAs retain their single-atom state without forming clusters after reactions. Conducting post-catalytic tests to study the morphology of the catalysts would address this gap in understanding?

Response to the reviewers' comments and changes made

Reviewer 1:

General comment: The authors present a very intriguing work of decomposing polyester wastes to diols via using Ru and Mo dual-atom catalysts supported by TiO₂. The topic is of great environmental implications and the authors have achieved desirable experimental/practical results. However, there are some problems to be addressed before the manuscript can be acceptable for publication in Nature Communications.

Comment 1: Why do the yields in Figure 1b fluctuate slightly with the number of Regeneration Cycle increasing instead of a descending trend?

Response 1: We sincerely thank you for this comment. In this work, the used catalyst was recovered after each reaction, and regenerated under an oxygen atmosphere at 200 °C that was used to prepare the fresh catalyst. The regenerated catalyst restored its original structure confirmed by the AC HADDF-STEM and XAS characterization (Supplementary Fig. 24), and remained catalytic activity comparable to the fresh catalyst. Consequently, the regenerated catalyst exhibits unchanged activity as illustrated in Fig. 1b, exhibiting excellent recyclability.

Comment 2: The authors points out that “the electron transfer from Mo to Ru sites”(in Line 302), i.e., Ru atoms act as electron acceptors, which seems to contradict with “The presence of the Mo atoms in Ru₄Mo₁/TiO₂ renders the Ru atoms to have a higher valence than that in Ru₆/TiO₂”(in Line 181-182),i.e., Ru atoms act as electron donators.

Response 2: We thank the reviewer for this valuable comment. In the as-prepared catalysts, e.g., Ru₄Mo₁/TiO₂, the electron transfer from Ru to Mo sites due to their electronic interaction, confirmed by XPS and XAS spectra, thus Ru SAs showing higher valence state compared to Ru clusters (in Line 181-182). However, in the case of lactic acid (LA) adsorption onto Ru₄Mo₁/TiO₂ (in Line 302), the calculated Bader

charges of Mo decrease from -2.20 to -2.27, and the charges of carbonyl O in LA increase from 1.07 to 1.15, while those of Ru increase from -1.49 to -1.48. These findings suggest that the adsorption of LA on the catalyst results in the redistribution of electrons, with transfer from Mo to carbonyl O site. We made relevant analysis and discussion in the revised manuscript.

Actually, your concerns belong to two different points, which do not contradict with each other. For clarity, we revised our corresponding statements.

Changes made: Furthermore, the Bader charges of the Ru and Mo in Ru-O-Mo before and after LA adsorption were calculated. The charges of Mo in Ru-O-Mo decrease from -2.20 to -2.27, and the charges of carbonyl O in LA increase from 1.07 to 1.15, while those of Ru only increase from -1.49 to -1.48. These findings suggest that the adsorption of LA at the Ru-O-Mo site results in the redistribution of electrons, with transfer from Mo to carbonyl O site. Consequently, the electron transfer pathway between Mo and O facilitates the activation of the C=O bond adsorbed at the Mo atom, thereby lowering the reaction energy barrier (page 10, line 17).

Comment 3: If “the Ru SA site has the sole capability for hydrogen activation” (in Line 318) is true, then it will be difficult to understand that “The Ru NPs in the catalyst can catalyze the hydrodeoxygenation of 1,2-propanediol to alkanes” (in Line 327). That is to say, where is 1,2-propanediol adsorbed on Ru NPs if not Ru atoms?

Response 3: We thank the reviewer for this comment. It is reasonable that the Ru SA site has different structure from Ru NPs confirmed by PDOS calculations (Supplementary Figs. 37). For H₂ and 1,2-propanediol adsorption, the calculations indicate that the Ru SA site shows strong capability to adsorb H₂, but little capability for 1,2-propanediol, which is thus ineffective for 1,2-propanediol hydrogenation; while Ru NPs have strong capability to adsorb H₂ and 1,2-propanediol simultaneously, which can explain why Ru NPs can achieve hydrodeoxygenation of 1,2-propanediol to alkanes. Based on the above discussion, we revised the statements on the Ru SA and Ru NPs.

Comment 4: In the sentence “where m_i is the mass of each product and m_0 represents

the feeding mass of plastics. ” (in Line 384), “mole” should be used instead of “mass”. The same is also true for “the overall mass of hydrogenation products” in Line 387.

Response 4: We cordially thank you for this valuable comment. We made a mistake in writing. We have changed " m_i " to " n_i ", " m_0 " to " n_0 ", " m_j " to " n_j " and "mass" to "mole" in the revised manuscript. Actually, we calculated the product yields and selectivity based on the moles of the collected products and the structural units of polyesters.

Revised 4: The yield of each product or byproduct, for example, the products and byproducts from PLA degradation including lactic acid, 1,2-propanediol, 1-propanol, isopropyl alcohol and propane was calculated by the following formula:

$$yield = \frac{n_i}{n_0} \times 100\%$$

where n_i is the moles of each product and n_0 represents the moles of the structural unit of feeding plastics.

The selectivity of each product or byproduct was calculated as follows:

$$selectivity = \frac{n_i}{\sum n_j} \times 100\%$$

where $\sum n_j$ is the overall moles of hydrogenation products (page 12, line 3).

Comment 5: The authors are encouraged to provide the adsorption energies of different adsorption configurations of LA on the Ru₄Mo₁/TiO₂ together with the configuration pictures in Supplementary Fig.29, Fig.30, Fig.34. and Fig.35.

Response 5: We thank you for the valuable suggestion. As suggested, we provide the adsorption energies of different adsorption configurations of LA on the Ru₄Mo₁/TiO₂ together with the configuration pictures in Supplementary Fig.35, Fig.36, Fig.40 and Fig.41.

Changes made:

Supplementary Fig. 35 DFT calculation-optimized configurations of LA on $\text{Ru}_4\text{Mo}_1/\text{TiO}_2$. Gray: Ru, purple: Mo, red: O, blue: Ti and pink: H
 The adsorption energies of the LA on the catalyst surface are (a) -0.59 eV, (b) -0.49 eV, (c) -0.89 eV, (d) -1.09 eV, (e) -1.12 eV, (f) -1.76 eV, respectively. It is indicated that LA is preferentially absorbed at the Mo site of $\text{Ru}_4\text{Mo}_1/\text{TiO}_2$.

Supplementary Fig. 36 DFT calculation-optimized configurations of LA adsorption at the Ru SA site. Gray: Ru, purple: Mo, red: O, blue: Ti and pink: H
 The adsorption energies of the LA on the different sites of the catalyst surface are (a) -1.30 eV (b) -1.38 eV, (c) -1.40 eV, respectively. It is indicated that LA forms hydrogen bonds with the lattice oxygen, but cannot bond effectively with Ru SA.

Supplementary Fig. 40 DFT calculation-optimized adsorption configurations of 1,2-propanediol and hydrogenated species on the $\text{Ru}_4\text{Mo}_1/\text{TiO}_2$. Gray: Ru, purple: Mo, red: O, blue: Ti and pink: H.

The adsorption energies of these three species on the catalyst surface are (a) -1.68 eV, (c) -2.16 eV, (e) -0.54 eV, and the adsorption energies of H_2 on Ru, Mo and Ti are -0.25 eV, -0.21 eV and -0.30 eV, respectively.

Supplementary Fig. 41 DFT calculation-optimized adsorption configurations of 1,2-propanediol and the hydrogenated species on Ru_6/TiO_2 . (a) $\text{C}_3\text{H}_6\text{-}2\text{OH}$, (b) $\text{C}_3\text{H}_6\text{-}2\text{OH}^*+\text{H}^*$, (c) $\text{C}_3\text{H}_6\text{-OH}^*$, (d) $\text{C}_3\text{H}_6\text{-OH}^*+\text{H}^*$, (e) $\text{C}_3\text{H}_6\text{-OH}$.

The adsorption energies of the three compounds on the catalyst surface are (a) -1.20 eV,

(c) -2.65 eV, (e) -0.68 eV, respectively.

Comment 6: The pictures mismatch letters captions in Supplementary Fig.33. (i.e., abcdef instead of adcebc).

Response 6: We thank the reviewer for the careful review. We made revision on the captions in Supplementary Fig.39 which was Fig.33 before.

Comment 7: The reaction energy barriers should be positive instead of negative, described below Supplementary Fig. 35.

Response 7: We appreciate you for the comment. In the original manuscript, we set the adsorption energy of C₃H₆-2OH at Ru-O-Mo site and Ru NP as 0, respectively, and calculated the reaction barrier of each step, which was negative. As you suggested, we rectified the original data to the relative values between two adjacent steps, which are energy barriers for the reactions between two adjacent steps. Obviously, the reaction barrier of C₃H₆-2OH*+H* → C₃H₆-OH*+H₂O over Ru₄Mo₁/TiO₂ is 0.66 eV, much higher than that obtained over Ru₆/TiO₂. This may explain why Ru₄Mo₁/TiO₂ is ineffective for the hydrogenation of 1,2-propanediol to propanol, while Ru₆/TiO₂ is effective.

Changes made: For the route C₃H₆-2OH → C₃H₆-2OH*+H* → C₃H₆-OH*+H₂O → C₃H₆-OH*+H* → C₃H₇-OH, the reaction energy barrier of each step over Ru₄Mo₁/TiO₂ and Ru₆/TiO₂ is -3.06, 0.66, -3.22, -0.54 eV and -3.65, 0.31, -3.54, 0.12 eV, respectively. (Figure S41).

Comment 8: The authors are encouraged to provide the adsorption energies and also its dissociation energies of H₂ on different metal atoms on the catalysts to further substantiate the conclusion that “the Ru SA sites activate H₂ to offer H for hydrogenation reactions”.

Response 8: We thank you for this valuable comment. As suggested, the adsorption energies and its dissociation energies of H₂ on different metal atoms on the catalysts were provided in the revised manuscript.

Changes made: The adsorption energies of H₂ on Ru, Mo and Ti were estimated to be -0.31, -0.21 and -0.30 eV, and the dissociation energies of H₂ on Ru, Mo and Ti were estimated to be 0.56, 2.51 and 3.74 eV, respectively (Figure S40).

Some language usage mistakes:

Comment 9: “X-ray adsorption fine structure (EXAFS)” in Line 194 should be “X-ray absorption fine structure (EXAFS)”.

Response 9: We appreciate the reviewer. We have corrected this mistake in the manuscript.

Comment 10: “Supplementary Fig. 20-S24” (in Line 210) should be “Supplementary Fig. 20-24”.

Response 10: We sincerely thank the reviewer, and corrected this mistake in the manuscript.

Reviewer 2:

General comment: The authors in this work have developed an interesting Ru-Mo dual-atom catalyst which demonstrates outstanding catalytic performance in converting various polyesters into diols. The structures and roles of Ru SA and Ru-O-Mo sites have been illustrated clearly. However, there are a few points that need to be addressed before this manuscript can be published.

Comment 1: In this manuscript, diols are produced via hydrolysis and subsequent hydrogenation. However, carboxylic acid is normally more difficult to be hydrogenated than ester, more details, such as NMR spectra of reaction mixtures containing various intermediate products, should be supplemented to prove this route. It is not clear enough what internal standard is and how the yields of intermediate and final products are exactly determined, thus Methods and Supporting Information need to be more detailed in analytical procedures and calculations.

Response 1: We thank you for this valuable comment. We agree with you that carboxylic acid is normally more difficult to be hydrogenated than ester. However, in the case of polyesters, the hydrolysis of polyester preferentially occurs compared to hydrogenation of polyester. As suggested, we provided the ^1H NMR spectra of the PLA decomposition solutions in the presence of H_2 over $\text{Ru}_4\text{Mo}_1/\text{TiO}_2$ in water for 6 and 9 h, as shown in Fig S7. It was indicated that lactic acid and 1,2-propanediol were obtained, without any other byproducts. For ^1H NMR analysis, paraformaldehyde was employed as the internal standard. The details for determining the amounts of intermediates and products were provided in the Methods and Supporting Information.

Changes made in the Method and Supporting Information: Paraformaldehyde was employed as the internal standard for the ^1H NMR analysis. The moles of paraformaldehyde were calculated based on its mass and molecular weight. The amounts of target products were determined by the following formula.

$$n_i = \frac{n_q \times a_i \times z_i}{6a_q}$$

where n_i is the mole of each product, n_q is the mole of paraformaldehyde, a_i is the ^1H NMR characteristic peak area of each product, a_q is the ^1H NMR peak area of paraformaldehyde, z_i is the H numbers of characteristic ^1H NMR peak for each product (page 11, line 43).

Supplementary Fig. 7 ^1H NMR spectra of the reaction solutions of PLA depolymerization in water in the presence of H_2 over $\text{Ru}_4\text{Mo}_1/\text{TiO}_2$ for 6 h (a) and 9 h (b). Conditions: PLA, 72 mg; $\text{Ru}_4\text{Mo}_1/\text{TiO}_2$, 10 mg; H_2O , 0.5 mL; H_2 , 4 MPa; 160 °C. Paraformaldehyde was employed as an internal standard for ^1H NMR analysis. The cumulative yields of LA and 1,2-propanediol reached up to 100% within 6 h, indicating complete depolymerization of PLA into monomers. As the reaction proceeded for 9 h, the yield of LA decreased, while the yield of 1,2-propanediol increased accordingly, without any other discernible byproducts. These results strongly suggest that 1,2-propanediol was accessed through hydrogenation of LA, and no side reaction occurred.

Comment 2: In Fig.2, the positions of b and c are disordered. In Fig.2d, it is not clear

how the two atoms in Area 1 or 2 are determined to be Ru and Mo, respectively, EDS mapping cannot present atomic-resolution.

Response 2: We sincerely thank you for the comments. We have corrected the positions of Fig. 2 (b) and (c), and replaced the previous AC HAADF-STEM and EDS images with clearer ones. In the AC HAADF-STEM images, the distinctive brightness of different elements allows for their differentiation. Given Ru's larger atomic number, it exhibits stronger brightness compared to Mo. By plotting the atomic brightness vs the vertical coordinate, we can effectively discern the presence of elements Ti, Mo, and Ru in the catalyst. In Figure 2d, Ru manifests higher peaks attributed to its elevated brightness, while Mo shows slightly lower peak. This visual representation provides a more intuitive depiction of the catalyst surface, elucidating the coexistence of Ru-Mo DA and Ru SA sites. This analytical method has been employed and documented in some scientific publications, such as references 27,35,36, i.e., DOI: 10.1021/jacs.3c07777, 10.1002/anie.202217449 and 10.1038/s41565-020-0665-x.

Changes made:

Fig. 2 Characterization of the catalysts. (a) TEM, (b) EDS mapping images and (c) AC HAADF-STEM image of Ru₄Mo₁/TiO₂. (d) The corresponding intensity

profile of the parent TiO₂ along the dashed rectangles, showing the isolated arrangement of Ru and Mo atoms. (e) AC HAADF-STEM and (f) EDS mapping images of Ru₆/TiO₂. (red: Ru, green: Mo, yellow: O, and cyan: Ti).

References:

27 Fu, N. et al. Controllable Conversion of Platinum Nanoparticles to Single Atoms in Pt/CeO₂ by Laser Ablation for Efficient CO Oxidation. *J. Am. Chem. Soc.* 145, 9540-9547, doi:10.1021/jacs.2c11739 (2023).

35 Zheng, X. et al. Dual-Atom Support Boosts Nickel-Catalyzed Urea Electrooxidation. *Angew. Chem. Int. Edit.* 62, e202217449, doi:https://doi.org/10.1002/anie.202217449 (2023).

36 Xiong, Y. et al. Single-atom Rh/N-doped carbon electrocatalyst for formic acid oxidation. *Nat. Nanotechnol.* 15, 390-397, doi:10.1038/s41565-020-0665-x (2020).

Comment 3: The sentence “.....of Ru₆/TiO₂ show that the Ru NPs with an average size of 3 nm are distributed on the surface of the TiO₂ support, without Ru SAs”: The STEM image cannot prove it and it is unreasonable that no Ru SA distributes on this catalyst at all.

Response 3: We sincerely thank you for this valuable comment. We agree with the reviewer that the STEM image cannot support the absence of the Ru SAs. In this catalyst Ru exists mainly in the form of nanoparticles, and trace amount of Ru SAs might exist. Therefore, we deleted "without Ru SAs" from that sentence.

Comment 4: In Figs.4c and d, more information should be added including the co-feed of H₂, catalyst and temperature.

Response 4: We sincerely thank you for the comment. As suggested, we have added related information in the revised manuscript.

Changes made: (c) Operando FTIR spectra for LA transformation over Ru₄Mo₁/TiO₂ and Mo₂/TiO₂ at room temperature and 120 °C in air. (d) Ex situ FTIR spectra for LA transformation over Ru₄Mo₁/TiO₂ for different time under optimal reaction conditions.

Comment 5: In the caption, “PLA 1 mmol” is not accurate, it should be corrected as “PLA 72 mg”.

Response 5: We sincerely thank you for this valuable comment. We made revision in the manuscript.

Comment 6: In Methods, the mass of products and feeding plastics is used to calculate yield and selectivity, but hydrogenated products are heavier than feeding plastics, for example, the mass of 1 mmol of propanediol is 76 mg, the mass of corresponding PLA is 72 mg. The yield and selectivity should be calculated based on the amount of carbon.

Response 6: We cordially thank you for this valuable comment. We made a mistake in writing. Actually, we calculated the product yields and selectivity based on the moles of the collected products and the structural units of polyesters. We have changed " m_i " to " n_i ", " m_0 " to " n_0 ", " m_j " to " n_j " and "mass" to "mole" in the revised manuscript.

Changes made: The conversion of polyester was calculated as follows:

$$conversion = \frac{m_0 - m_s}{m_0} \times 100\%$$

where m_0 is the feeding mass of polyester and m_s is the weight of solid residue after reaction.

The yield of each product or byproduct, for example, the products and byproducts from PLA degradation including lactic acid, 1,2-propanediol, 1-propanol, isopropyl alcohol, propane, was calculated by the following formula:

$$yield = \frac{n_i}{n_0} \times 100\%$$

where n_i is the moles of each product and n_0 represents the moles of the structural unit of feeding plastics.

The selectivity of each product or byproduct was calculated as follows:

$$selectivity = \frac{n_i}{\sum n_j} \times 100\%$$

where $\sum n_j$ is the overall moles of hydrogenation products.

Reviewer 3:

General comment: In this manuscript, the authors demonstrated the efficient recycling of polyester wastes to diols using Ru and Mo dual atoms catalysis. This work is definitely a significant with respect to current global challenges and will have remarkable impact in the field. However, there are certain points to be addressed in a revision in order that this manuscript may fulfils the high standards of Nature Communications.

Comment 1: The authors have used a combination of Ru and Mo, as Ru₄Mo₁ is the major catalyst among other combinations tried for Better conversion. It would be beneficial for the readers to understand the rationale behind this preference.

Response 1: We appreciate you for the comments. In this work, the degradation of polyesters into diols undergoes hydrolysis of polyesters to dicarboxylic acids and diols or hydroxyl carboxylic acids depending on the structures of the polymers, and subsequent hydrogenation of carboxylic acids into diols. Therefore, the catalysts are required to be capable of catalyzing the hydrolysis of polyesters and the subsequent hydrogenation of carboxylic acid exclusively to diols. Since the TiO₂ support of the as-prepared catalysts could achieve the hydrolysis of polyesters, the catalysts that could achieve hydrogenation of carboxylic acid exclusively to diols without hydrodeoxygenation of diols are needed. Among the resultant catalysts, Ru₄Mo₁/TiO₂ meets the above requirements and exhibits the highest activity. We discussed the rationale for selecting catalyst for polyester degradation into diols in the revised manuscript as you suggested.

Changes made: Considering the macromolecular solid nature of polyester, a deliberate decision was made to implement a hydrolysis step as a means to lower the overall reaction temperature. This approach effectively facilitates the degradation of the polyester to dicarboxylic acids and diols or hydroxyl carboxylic acids depending on the structures of the polymers, and subsequent hydrogenation of carboxylic acids into diols. Therefore, the catalysts are required to be capable of catalyzing the hydrolysis of polyesters and the subsequent hydrogenation of carboxylic acid exclusively to diols without hydrodeoxygenation of diols (page2, line 35).

Comment 2: Did the authors try a blank reaction with TiO₂ to elaborate more on the role of TiO₂ nanoparticles? How is it different when compared to TiO₂ as support?

Response 2: Actually, we performed the degradation of PLA utilizing TiO₂ as the catalyst under optimized conditions, and hydrolysis of PLA occurred to produce lactic acid in a yield of 100%. This suggests that the TiO₂ support of the as-prepared catalysts could catalyze the hydrolysis of PLA to lactic acid. Relevant information was provided in Figure 1a.

Fig. 1. (a) Catalyst activity for PLA degradation.

Comment 3: While the authors claim the formation of DAs and SAs, the evidence provided by HAADF, STEM, and EDS spectra is not entirely convincing. Few other characterizations are recommended to support the fact. For e.g., STM analysis for further validating the fact that DAs and SAs were formed in the reaction.

Response 3: We thank you for this valuable comment. For our catalyst samples, it is difficult to observe the single atoms by STM and AFM due to their irregular nanoparticle structures. Instead, we employed AC HAADF-STEM to observe the catalyst from multiple angles and locations, and more AC HAADF-STEM and EDS mapping-line-scan images were provided in the revised Supporting Information, which support the existence of both DAs and SAs.

Changes made: To provide a more comprehensive illustration of the formation of Ru-O-Mo DAs and Ru SAs, we employed AC HAADF-STEM to observe Ru₄Mo₁/TiO₂ from multiple angles and locations. The obtained images and EDS line-scan analysis (Supplementary Figs. 12-14) provide solid evidences of the coexistence of Ru SAs and Ru-Mo DAs (page 4, line 48).

Supplementary Fig. 12 AC HAADF-STEM image and EDS mapping of Ru₄Mo₁/TiO₂.

Supplementary Fig. 13 EDS line-scan images of Ru₄Mo₁/TiO₂.

The line-scan results show that Ru SAs and Ru-Mo DAs sites are present on the surface of the catalysts.

Supplementary Fig. 14 AC HAADF-STEM images of Ru₄Mo₁/TiO₂.

Comment 4: Exploring the reactivity when using Ru nanoparticles instead of single atoms would provide valuable insights into the catalytic mechanism.

Answer: We thank you for this valuable comment. Actually, in this work Ru₆/TiO₂ and Ru₄Mo_{0.1}/TiO₂ were used as control catalysts with Ru nanoparticles (NPs) to compare their activities for hydrogenation of propylene glycol in water. It was indicated that the presence of Ru NPs led to hydrogenation of propylene glycol to produce monohydric alcohols or alkanes. The relevant results are shown in Figs. S2 and S3.

Comment 5: Although the catalyst exhibits excellent recyclability, it remains unclear how SAs retain their single-atom state without forming clusters after reactions. Conducting post-catalytic tests to study the morphology of the catalysts would address this gap in understanding?

Answer: We appreciate you for the comments. Actually, the microscopic characterization of the used catalyst indicated that the Ru SAs aggregated into clusters after the hydrogenation reaction. However, upon treating the used catalyst under the oxygen condition that was used to prepare the fresh catalyst, the formed Ru clusters became into SAs again confirmed by AC HAADF-STEM and XAS characterization (Supplementary Fig. 24). The as-regenerated catalyst showed the similar activity to the fresh one. That is, this catalyst could be easily regenerated remaining its original activity, exhibiting excellent recyclability.

Response 5: XAS characterization on the used and regenerated catalysts was conducted to complement the electronic microscopy findings (Supplementary Fig. 24 and 25). Consistent with the AC HAADF-STEM results, the XAS analysis in R-space revealed the presence of Ru clusters and Mo SAs in the used catalyst. However, in the regenerated catalyst no agglomerated Ru clusters were observed, and SAs instead were present. This means that the regeneration process of the used catalyst led to the formation of SAs once again (page 6, line 30).

Supplementary Fig. 24 Ru K-edge R-space X-ray absorption near-edge structure spectra of (a) used and (b) regenerated catalyst.

Supplementary Fig. 25 Mo K-edge R-space X-ray absorption near-edge structure spectrum of used catalyst.

REVIEWER COMMENTS

Reviewer #1 (Remarks to the Author):

The authors have revised their manuscript comprehensively. Most of the problems raised have been addressed. However, there are still some problems to be further addressed to guarantee its final quality for publication in Nature Communication.

Comment 2 (continued): Why are the charges of carbonyl O positive, while those of Mo and Ru are negative? The authors state that “with transfer from Mo to carbonyl O site”, which means Mo acts as the electron donator. However, “The charges of Mo in Ru-O-Mo decrease from -2.20 to -2.27”, which means Mo acts as the electron acceptor. The authors should think carefully of the relationship between the electron transfer and the charges of the atoms concerned.

Comment 7(continued): If the reaction energy barrier of a reaction is negative, then the energy of transition state is lower than that of the initial state, i.e., it is probably a spontaneous reaction. Or rather, 3 of the 4 elementary reactions have negative reaction energy barriers, meaning they are spontaneous. There may be something wrong in the calculations of energetics.

Reviewer #2 (Remarks to the Author):

The comments have been fully addressed and the manuscript is significantly improved. The manuscript can be accepted as it is.

Reviewer #3 (Remarks to the Author):

The authors have effectively addressed the concerns raised in this manuscript. Based on their satisfactory responses, I recommend that this article be considered for publication in Nature Communications.

Response to the reviewers' comments and changes made

Reviewer 1:

General comment: The authors have revised their manuscript comprehensively. Most of the problems raised have been addressed. However, there are still some problems to be further addressed to guarantee its final quality for publication in Nature Communication.

Comment 2 (continued): Why are the charges of carbonyl O positive, while those of Mo and Ru are negative? The authors state that “with transfer from Mo to carbonyl O site”, which means Mo acts as the electron donator. However, “The charges of Mo in Ru-O-Mo decrease from -2.20 to -2.27”, which means Mo acts as the electron acceptor. The authors should think carefully of the relationship between the electron transfer and the charges of the atoms concerned.

Response 2: We thank you for this valuable comment. To analyze the electron transfer between the catalyst and the reactant, we calculated Bader charges of the Mo and Ru atoms in Mo-O-Ru and carbonyl O atom of lactic acid that interacts with the catalyst. Based on Richard Bader's definition, a negative value of Bader charges represents a loss of electrons and a positive one represents a gain of electrons. In this work, "The charges of Mo in Ru-O-Mo decrease from -2.20 to -2.27", represents that Mo loses electrons, changing from 2.20 to 2.27 electrons; "the charges of carbonyl O in LA increase from 1.07 to 1.15", represents the change of O from gaining 1.07 electrons to 1.15 electrons. Since Mo loses electrons and carbonyl O gains electrons, Mo is an electron donor and O is an electron acceptor. That is, electrons are transferred from Mo to O atoms. For clarity, we rewrote the paragraph for analysis on Bader charges of Mo and Ru in Mo-O-Ru before and after lactic acid adsorption in the revised manuscript as follows.

Changes made: Furthermore, the Bader charges of the Ru and Mo in Ru-O-Mo before and after LA adsorption were calculated. The Bader charges of Mo in Ru-O-Mo

decrease from -2.20 to -2.27, while those of carbonyl O in LA increase from 1.07 to 1.15 as LA is adsorbed on Mo-O-Ru. This suggests that the adsorption of LA at the Mo-O-Ru site makes the Mo atom lose more electrons, along with carbonyl O in LA gaining more electrons. Notably, the Bader charges of Ru only increase from -1.49 to -1.48, suggesting little influence on its electronic structure after LA adsorption. The above findings demonstrate that the adsorption of LA at the Ru-O-Mo site results in the redistribution of electrons, with electron transfer from Mo to carbonyl O site. Consequently, the electron transfer pathway between Mo and O facilitates the activation of the C=O bond adsorbed at the Mo atom, thereby lowering the reaction energy barrier.

Comment 7 (continued): If the reaction energy barrier of a reaction is negative, then the energy of transition state is lower than that of the initial state, i.e., it is probably a spontaneous reaction. Or rather, 3 of the 4 elementary reactions have negative reaction energy barriers, meaning they are spontaneous. There may be something wrong in the calculations of energetics.

Response 7: We appreciate you for the comment. In the original Fig. 5e, we provided the relative energies between two steps for 1,2-propanediol hydrogenation to isopropanol over Ru₄Mo₁/TiO₂ and Ru₆/TiO₂, respectively. For the elementary reaction $C_3H_6-2OH \xrightarrow{H^*} C_3H_6-2OH^*+H^*$ and $C_3H_6-OH^*+H_2O \xrightarrow[-H_2O]{H^*} C_3H_6-OH^*+H^*$, we omitted the energy of H₂ as a reactant before, resulting in a non-conservation of the materials around the reaction, thus making the difference of reaction free energy negative. In the revised Fig. 5e, the energy of 1/2 H₂ was included to make the material conserved, and the reaction energy barrier of each step was obtained as illustrated in Fig. 5e. Therefore, the reaction energy barriers are positive and plausible.

Changes made:

Fig. 5 (e) Free energy diagrams of 1,2-propanediol hydrogenation on $\text{Ru}_4\text{Mo}_1/\text{TiO}_2$ and Ru_6/TiO_2 .

Fig. 5e illustrates the energy barrier of each step for hydrogenation of 1,2-propanediol to isopropanol over different catalysts. Obviously, the step $\text{C}_3\text{H}_6\text{-2OH}^* + \text{H}^* \rightarrow \text{C}_3\text{H}_6\text{-OH}^* + \text{H}_2\text{O}$ over $\text{Ru}_4\text{Mo}_1/\text{TiO}_2$ has a higher energy barrier (0.66 eV), making it an inefficient hydrogenation pathway.

Supplementary Fig. 41

For the route $\text{C}_3\text{H}_6\text{-2OH} \xrightarrow{\text{H}^*} \text{C}_3\text{H}_6\text{-2OH}^* + \text{H}^* \rightarrow \text{C}_3\text{H}_6\text{-OH}^* + \text{H}_2\text{O} \xrightarrow{\text{H}^*} \text{C}_3\text{H}_6\text{-OH}^* + \text{H}^* \rightarrow \text{C}_3\text{H}_7\text{-OH}$, the reaction energy barrier of each step over $\text{Ru}_4\text{Mo}_1/\text{TiO}_2$ and Ru_6/TiO_2 is 0.33, 0.66, 0.16, -0.54 eV and -0.27, 0.31, -0.16, 0.12 eV, respectively.

Reviewer 2:

General comment: The comments have been fully addressed and the manuscript is significantly improved. The manuscript can be accepted as it is.

Reviewer 3:

General comment: The authors have effectively addressed the concerns raised in this manuscript. Based on their satisfactory responses, I recommend that this article be considered for publication in Nature Communications.

REVIEWERS' COMMENTS

Reviewer #1 (Remarks to the Author):

All of the problems raised have been addressed soundly. The manuscript can be considered for publication in Nature Communications.